**Data Availability Statement:** All relevant data are within the paper and its Supporting information files.

# Differentially expressed microRNAs during the differentiation of muscle-derived stem cells into insulin-producing cells, a promoting role of microRNA-708-5p/STK4 axis

**Yu Ren**[1☯], **Xiao Wang**[2☯], **Hongyu Liang**[1], **Yuzhen Ma**[🆔][3]*

**1** Scientific Research Department, Inner Mongolia People's Hospital, Hohhot, Inner Mongolia Autonomous Region, China, **2** Clinical Medical Research Center, Inner Mongolia People's Hospital, Hohhot, Inner Mongolia Autonomous Region, China, **3** Reproductive Medicine Center, Inner Mongolia People's Hospital, Hohhot, Inner Mongolia Autonomous Region, China

☯ These authors contributed equally to this work.

* xingshengzhao2021@163.com

## Abstract

### Objective

Stem cell therapy is a promising approach for diabetes via promoting the differentiation of insulin-producing cells (IPCs). This study aimed to screen the differentially expressed miR-NAs (DEmiRNAs) during the differentiation of muscle-derived stem cells (MDSCs) into IPCs, and uncover the underlying function and mechanism of a specific DEmiRNA, miR-708-5p.

### Methods

MDSCs were successfully isolated from the leg muscle of rats, and were induced for IPCs differentiation through a five-stage protocol. miRNA microarray assay was performed for screening DEmiRNAs during differentiation. The features of MDSCs-derived IPCs were identified by qRT-PCR, flow cytometry, and immunofluorescence staining. The targeting of STK4 by miR-708-5p was examined by luciferase assay. The protein expression of STK4, YAP1, and p-YAP1 was determined by Western blot and immunofluorescence staining.

### Results

MDSCs were successfully isolated and differentiated into IPCs. A total of 12 common DEmiRNAs were obtained during five-stage differentiation. Among them, miR-708-5p that highly expressed in MDSCs-derived IPCs was selected. Overexpression of miR-708-5p upregulated some key transcription factors (Pdx1, Ngn3, Nkx2.2, Nkx6.1, Gata4, Gata6, Pax4, and Pax6) involving in IPCs differentiation, and increased insulin positive cells. In addition, STK4 was identified as the target gene of miR-708-5p. miR-708-5p overexpression downregulated the expression of STK4 and the downstream phosphorylated YAP1.

**Funding:** This work was supported by Inner Mongolia Autonomous Region Science and Technology Innovation Guidance Project [No. KCBJ2018042] and Natural Science Foundation of Inner Mongolia Autonomous Region [No. 2016BS0314], funder play the role of Conceptualization, Data curation, Writing – original draft.

**Competing interests:** The authors have declared that no competing interests exist.

## Conclusions

There were 12 DEmiRNAs involved in the differentiation of MDSCs into IPCs. miR-708-5p promoted MDSCs differentiation into IPCs probably by targeting STK4-mediated Hippo-YAP1 signaling pathway.

## Introduction

Diabetes is a common metabolic disease defined as the disorder of homeostatic control of blood sugar levels, which mainly caused by the insulin secretion deficiency resulting from the destruction of pancreatic β-cells [1]. The incidence of diabetes is progressively increasing annually worldwide, posing a great threat to people's health. Recently, the development of insulin-producing pancreatic β-cells from stem cells has become a potential approach to treat diabetes [2]. However, the therapeutic strategy of stem cell-derived insulin-producing cells (IPCs) is technically immature and the underlying mechanisms are still illusive.

Muscle-derived stem cells (MDSCs) are a type of postnatal adult stem cells, which possess a high regeneration and differentiation capacity [3]. Currently, MDSCs are increasingly popular used in stem cell therapy, due to abundant and easily accessible source and strong regenerative capacity [4]. MDSCs are considered to have the capacity of differentiation into muscular, vascular, nerve, bone, and cardiac lineage cells [5]. Mitutsova et al. indicated that MDSCs can differentiate into IPCs in pancreatic islets [6]. Lan et al. [7] also confirmed that muscle-derived stem/progenitor cells are capable of differentiating into insulin-producing clusters. However, the underlying mechanisms of MDSCs differentiation into IPCs have not been fully characterized.

MicroRNAs (miRNAs) are endogenous small-non-coding RNAs, which act as crucial regulatory roles in gene expression by complementary binding to the 3'-untranslated regions (3'-UTR) of target mRNAs [8]. Numerous evidences have supported that miRNAs are involved in the differentiation of various stem cells into IPCs by regulating target genes [9–11]. For instance, miRNA-375 promoted the differentiation of human embryonic stem cells into IPCs by affecting its target genes and pancreatic development-related genes, including PDPK1, INSM1, HNF1B, GATA6, NOTCH2, PAX6, and CADM1 [9]. miRNA-690 facilitated the differentiation of induced pluripotent stem cells into IPCs by targeting Sox9 [11]. In addition, some miRNAs, such as miRNA-375, miRNA-29, and miRNA-7, are capable of regulating the expression of key functional genes in pancreatic β-cells [12, 13]. However, there still some miRNAs involved with MDSCs-derived IPCs differentiation have not been illustrated.

In this study, differentially expressed miRNAs (DEmiRNAs) associated with IPCs differentiation from MDSCs were screened via miRNA microarray analysis, and miRNA-708-5p (miR-708-5p) was selected for further functional analyses. The target genes of miR-708-5p were predicted and the underlying mechanism of miR-708-5p regulating the differentiation of MDSCs into IPCs was uncovered. These findings shed light on the underlying mechanism of MDSCs-derived IPCs differentiation and provide a novel therapeutic strategy for diabetes.

## Methods

### MDSCs isolation, culture and identification

MDSCs were isolated from the leg muscle of Wistar rats according to the previously described method [14]. Briefly, the muscle tissues were collected from Wistar rats and digested with 0.1% type I collagenase for 1 h at 37˚C. After centrifugation for 5 min at 150 ×$g$, the pellet was

resuspended and cultured in Dulbecco's modified Eagle's medium (DMEM)/F12 containing 20% fetal bovine serum (FBS, Gibco, MA, USA), 10% horse serum, and 1% penicillin/streptomycin. After cultured for 10 h and 48 h, the morphology of cells was observed under a fluorescence microscope (Olympus, Japan). This study was approved by the Institutional Animal Care and Use Committee of Inner Mongolia People's Hospital.

### *In vitro* differentiation of MDSCs into IPCs

MDSCs were induced to differentiate into IPCs via a five-stage differentiation method according previous description with slight modification [15]. At stage 1, MDSCs were cultured in the Roswell Park Memorial Institute (RPMI) medium containing 0.2% FBS (Gibco, MA, USA) and 100 ng/mL Activin A for two days. At stage 2, cells were cultured in RPMI medium containing 2% FBS and 25 ng/mL keratinocyte growth factor (KGF) for three days. At stage 3, cells were cultured in DMEM (Gibco, MA, USA) containing 1% B-27, 2 μM retinoic acid, 0.25 μM cyclopamine, and 50 ng/mL Noggin for three days. At stage 4, cells were cultured in Iscove's Modified Dulbecco's Medium (IMDM) containing 1% B-27, 10 mM nicotinamide, and 100 nM GLP-1 (Preproglucagon 72–107 amide) for three days. At stage 5, cells were cultured in IMDM containing 1% B-27, 10 mM nicotinamide, 100 nM GLP-1, 50 ng/mL insulin-like growth factor 1 (IGF-I), and 50 ng/mL human hepatocyte growth factor (HGF) for three days. At each stage, cells were visualized using a fluorescence microscope (Olympus, Japan). After 15 days of induction, the MDSCs-derived IPCs were obtained for further experiments. MDSCs at fifth-generation that cultured in RPMI media containing 25 ng/mL Wnt-3a and 100 ng/mL Activin A were used as the control.

### miRNA microarray assay and bioinformatics analysis

Total RNA was extracted from MDSCs-derived IPCs at five stages using TRIZOL reagent (AidLab, China). A miRNA library was constructed using the Illumina TruSeq Small RNA kit (Illumina, CA, USA), and miRNA-seq was performed on the Illumina Hiseq 2500 platform. The read quality was evaluated using the online tool FastQC v0.11.9 (https://www.bioinformatics.babraham.ac.uk/projects/fastqc/). The high-quality sequencing data (clean reads) were screened according to the criteria as follows: 1) Remove the 3' linker sequence in the reads, and remove the reads without insert fragments due to the self-ligation of the linker; 2) Remove the reads with low sequencing quality in 3'-base (the quality value is less than 20); 3) Remove reads containing unknown base N; 4) Choose reads with length between 18nt and 32nt. These analysis parameters have been added to the Methods section. The obtained clean reads were applied for DEmiRNAs identification using DESeq2 (v1.34.0) [16], DEGseq (v1.48.0) [17] and edgeR (v3.36.0) [18] in Bioconductor. miRanda (v3.3a), TargetScan (http://www.targetscan.org/vert_72/), and RNAhybrid [19] were employed to determine target genes of DEmiRNAs. For functional enrichment analyses, the target genes of DEmiRNAs were annotated through Gene Ontology (GO) and Kyoto Encyclopedia of Genes and Genomes (KEGG) pathway analyses.

### Cell transfection

The miR-708-5p mimics, mimics negative control (NC), STK4 siRNA (siSTK4), and siRNA negative control (siNC) were purchased from Genepharm company (Genepharm, Shanghai, China). These agents were transfected into MDSCs using Lipofectamine 3000 (Invitrogen, CA, USA) for 48 h. The transfected cells were induced for differentiation into IPCs in subsequent experiments.

**Table 1. The primers for qRT-PCR.**

| Genes | Primer sequences (5' to 3') |
|---|---|
| Pdx1 | Forward: CCTTTCCCGAATGGAACCGA<br>Reverse: TTTTCCACGCGTGAGCTTTG |
| NGN3 | Forward: CATAGCGGACCACAGCTTCT<br>Reverse: GGCTACCAGCTTGGGAAACT |
| Nkx2.2 | Forward: ACCAACACAAAGACGGGGTT<br>Reverse: ACCAGATCTTGACCTGCGTG |
| Nkx6.1 | Forward: GCGGACCAAGTGGAGAAAGA<br>Reverse: AGTCTCCGAGTCCTGCTTCT |
| Gata4 | Forward: ATGGGTCCTCCATCCATCCA<br>Reverse: GCTGTTCCAAGAGTCCTGCT |
| Gata6 | Forward: TCCTCTTCCTCCTCCTGCTC<br>Reverse: GAAGAGCAACAGGTCCTCCC |
| Pax4 | Forward: GACGGTCTCAGCAGTGTGAA<br>Reverse: GGGGACTAGGAAGAGCTGGA |
| Pax6 | Forward: CAGAACAGTCACAGCGGAGT<br>Reverse: CAGACCCCCTCGGAGAGTAA |
| GAPDH | Forward: GCAAGGATACTGAGAGCAAGAG<br>Reverse: GGATGGAATTGTGAGGGAGATG |
| U6 | Forward: AACTCTGGCGTGGATTACCG<br>Reverse: CACATCCGCCTCTTCTGTGT |

## Quantitative RT-PCR

Total RNA was isolated from MDSCs-derived IPCs using TRIZOL reagent (AidLab, China), and reverse transcription reaction was conducted using a FastKing OneStep Probe RT-qPCR MasterMix (TIANGEN, China). qRT-PCR was performed in Mx3000P Real-Time PCR System (Stratagene, CA, USA) under the following reaction program: 95˚C for 3 min, 40 cycles of 95˚C for 12 s and 62˚C for 40 s. The relative mRNA expression level of miRNA and IPCs-related genes were calculated using the $-2^{\Delta\Delta Ct}$ method. U6 was used as a reference gene for miRNA and glyceraldehyde-3-phosphate dehydrogenase (GAPDH) for IPCs-related genes. The primers used in this study are listed in Table 1.

## Flow cytometry

For identification of the insulin positive cells, resuspended MDSCs-derived IPCs ($1\times10^6$ cells/mL) were incubated with the anti-h/b/m insulin APC-conjugated rat IgG2A (R&D Systems, MN, USA) for 30 min at room temperature in the dark. The insulin positive cells were detected using a flow cytometry (Beckman Coulter, Germany) and analyzed using CellQuest software (BD Biosciences, NJ, USA).

## Dual-luciferase reporter assay

The binding site of miR-708-5p on STK4 was predicted using Starbase (v3.0; http://starbase. sysu.edu.cn/index.php). The luciferase reporter assay was performed to observe the interaction between miR-708-5p and STK4. The mutant STK4 (STK4-MT) was established via mutating the putative binding site of miR-708-5p in STK4 3'-UTR. Wild-type STK4 (STK4-WT) and STK4-MT were cloned into pGL3 alkaline luciferase vector. pGL3-STK4-WT and pGL3-STK4-MT were respectively co-transfected with miR-708-5p mimics into HEK293T cells. At 48 h post-transfection, Firefly and Renilla (Firefly: Renilla = 1: 0.1) luciferase activities

were measured using a Dual-Luciferase Reporter Assay System (Promega, Madison, USA) according to the manufacturer's instruction.

### Immunofluorescence staining

MDSCs-derived IPCs were washed with phosphate buffered saline (PBS) and then fixed with 3% paraformaldehyde for 15 min at room temperature. After washed with PBS three times, cells were permeabilized with 1% Triton X-100 for 10 min at room temperature, followed by blocking with 3% bovine serum albumin (BSA) for 30 min. Then, cells were incubated with primary antibodies (1:1,000) at 4°C overnight, followed by incubating with fluorescence secondary antibody (1:500, Abcam, UK) and DAPI (Solarbio, China) for 1 h at room temperature in the dark. Subsequently, cells were observed using a confocal laser scanning microscope (Olympus, Japan). Primary antibodies are listed as follows: anti-insulin antibody, anti-C-peptide antibody, anti-Pdx1 antibody, anti-Nkx6.1 antibody, anti-STK4 antibody, and anti-p-YAP1 antibody (Abcam, UK).

### Western blotting

MDSCs-derived IPCs were lysed with a RIPA lysis buffer (Takara Bio, Japan) for 15 min at 4°C to extract total protein. Protein concentrations were measured using a BCA Protein Assay Kit (Thermo Fisher Scientific, CA, USA). Total proteins were separated by SDS-PAGE and then transferred onto polyvinylidene difluoride (PVDF) membranes. Membranes were incubated with blocking buffer (5% nonfat dry milk dissolved in 1× Tris buffered saline with 0.1% tween-20 (1× TBST)) at room temperature for 1 h, followed by the primary antibodies (STK4, YAP1, p-YAP1, and GAPDH (1:1,000, Abcam, UK) at 4°C overnight. Then, membranes were incubated with horseradish peroxidase (HRP)-conjugated goat anti-rabbit IgG secondary antibody (1:500, MultiSciences, China) for 1 h at room temperature. Protein bands were visualized using ECL reagent kit (Thermo Fisher Scientific, CA, USA). Protein images were captured using a ChemiDoc™ imaging system (Bio-Rad, CA, USA). GAPDH was used as a reference control.

### Statistical analysis

All the experiments were performed with three independent repetitions. Data were expressed as the mean ± standard deviation (SD). Significant differences between different groups were determined via one-way of analyses variance (ANOVA), followed by Tukey's test. Statistical analysis was carried out using SPSS 27.0 software (IBM, IL, USA). $P < 0.05$ was considered significant differences.

## Results

### The isolation and identification of MDSCs

MDSCs were isolated from the muscle tissues from the leg of rats, which showed circular morphology (Fig 1A). After induced for 10 h, MDSCs presented as fusiform or polygonal morphology without complete cell adherence (Fig 1B). After 48 h, the fusiform or polygonal MDSCs were expanded and completed adherence (Fig 1C). In addition, the muscle cells-specific proteins, including desmin, sarcomeric α-actinin, MyoD1, Myf5, and Pax7, presented positive expression in MDSCs compared with 0 h MDSCs (Fig 1D–1I).

### *In vitro* differentiation of MDSCs-derived IPCs

The differentiation process of MDSCs-derived IPCs were divided into five stages. From stage 1 to 5, the cell morphology gradually changed from fusiform or polygonal to spherical (Fig 2A).

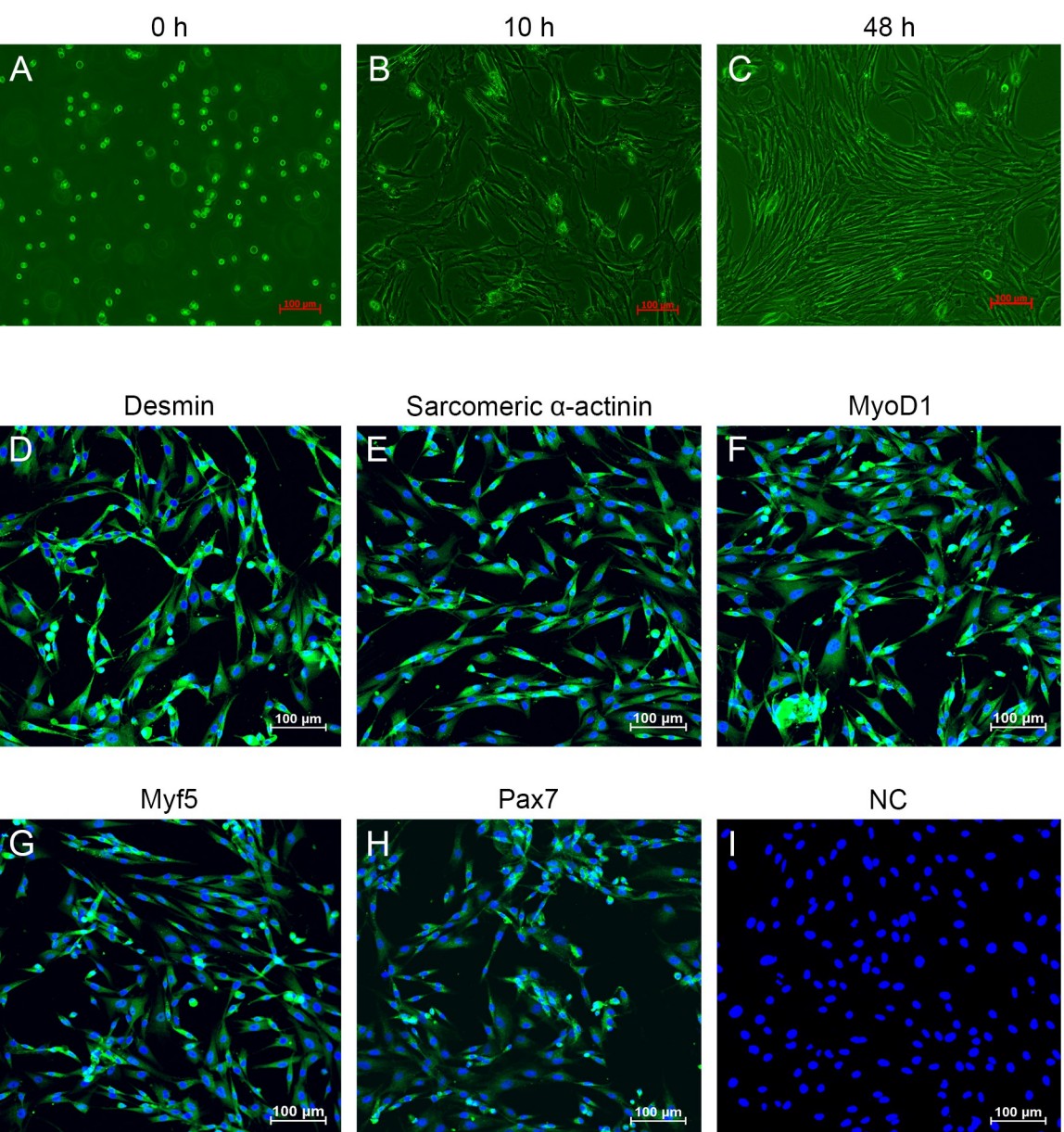

**Fig 1. The isolation and identification of muscle-derived stem cells (MDSCs).** (A) MDSCs were isolated from the leg muscle of Wistar rats (0 h). (B) MDSCs were induced for 10 h. (C) MDSCs were induced for 48 h. (D-H) The MDSCs markers, including desmin, sarcomeric α-actinin, MyoD1, Myf5, and Pax7, were detected using immunofluorescence staining, respectively. (I) MDSCs stained with DAPI were considered as the negative control (NC). Scale bar is 100 μm.

Insulin is a hormone produced by IPCs and C-peptide is an active form of insulin. These two markers (insulin and C-peptide) of IPCs were detected to evaluate the features of MDSCs-derived IPCs. Immunofluorescence staining showed that most of MDSCs-derived IPCs were positive for insulin and C-peptide compared with MDSCs (Fig 2B). Moreover, the flow cytometry presented that the percentage of insulin in MDSCs-derived IPCs was significantly higher than that in the control cells ($P < 0.001$) (Fig 2C).

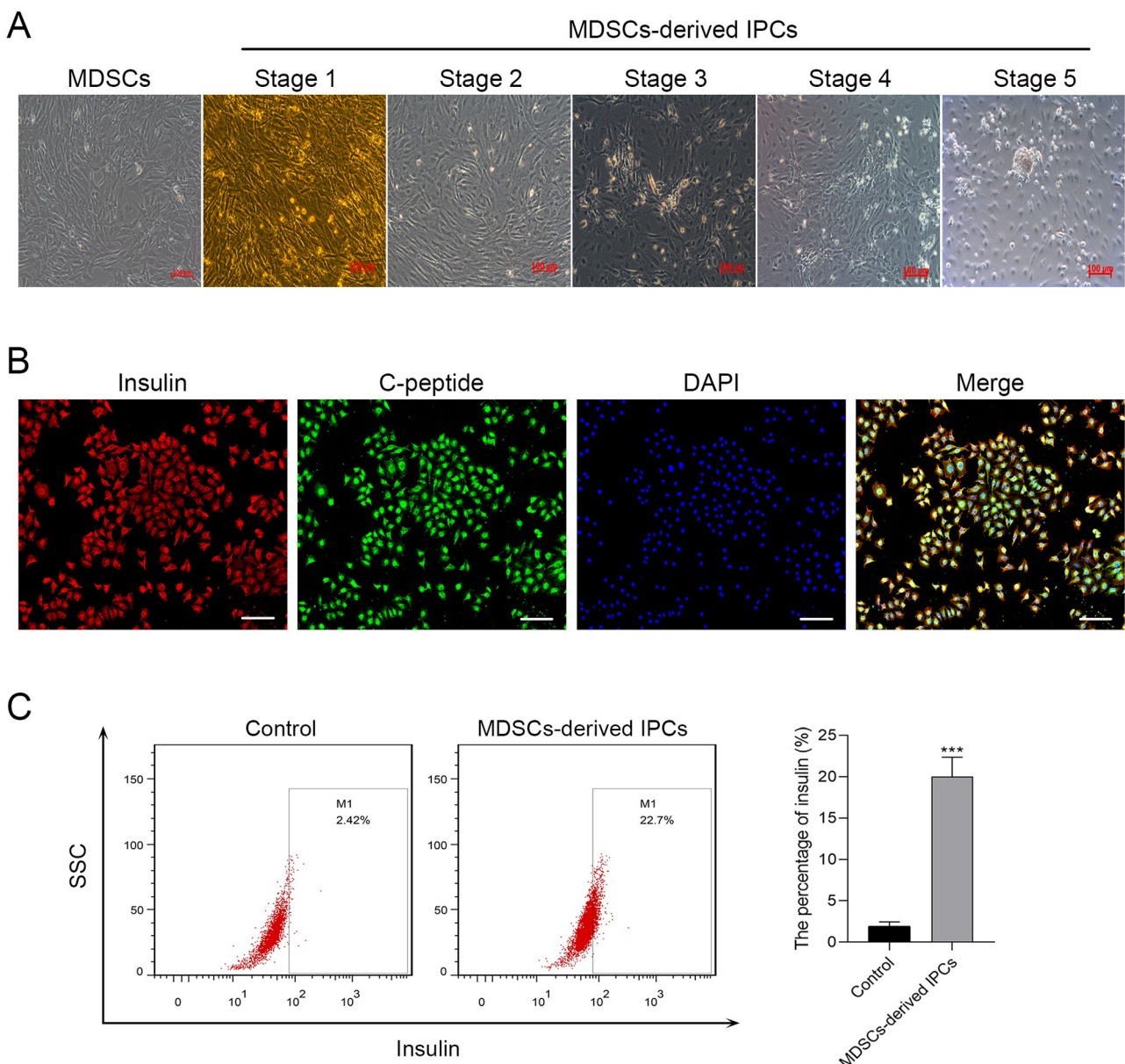

**Fig 2. The differentiation of MDSCs into IPCs *in vitro*.** (A) Morphologies of MDSCs-derived IPCs during stage 1–5 of differentiation. Scale bar is 100 μm. (B) MDSCs-derived IPCs at stage 5 were identified via co-immunostaining of insulin (red) and C-peptide (green). Nuclear DAPI staining was presented in blue. Scale bar is 200 μm. (C) The percentage of insulin in MDSCs-derived IPCs was measured using flow cytometry. ***$P < 0.001$ compared with the Control.

## miRNA profiling during MDSCs differentiation into IPCs

miRNA microarray assay was performed to screen the DEmiRNAs associated with the differentiation of MDSCs into IPCs. A Venn diagram showed that there were 49, 77, 119, 159, and 204 DEmiRNAs existing in differentiation stage 1–5, respectively. A total of 12 common DEmiRNAs were obtained in the five stages of differentiation (Fig 3A). The expression of the common DEmiRNAs at five stages of differentiation was clustered by a heatmap (Fig 3B). As

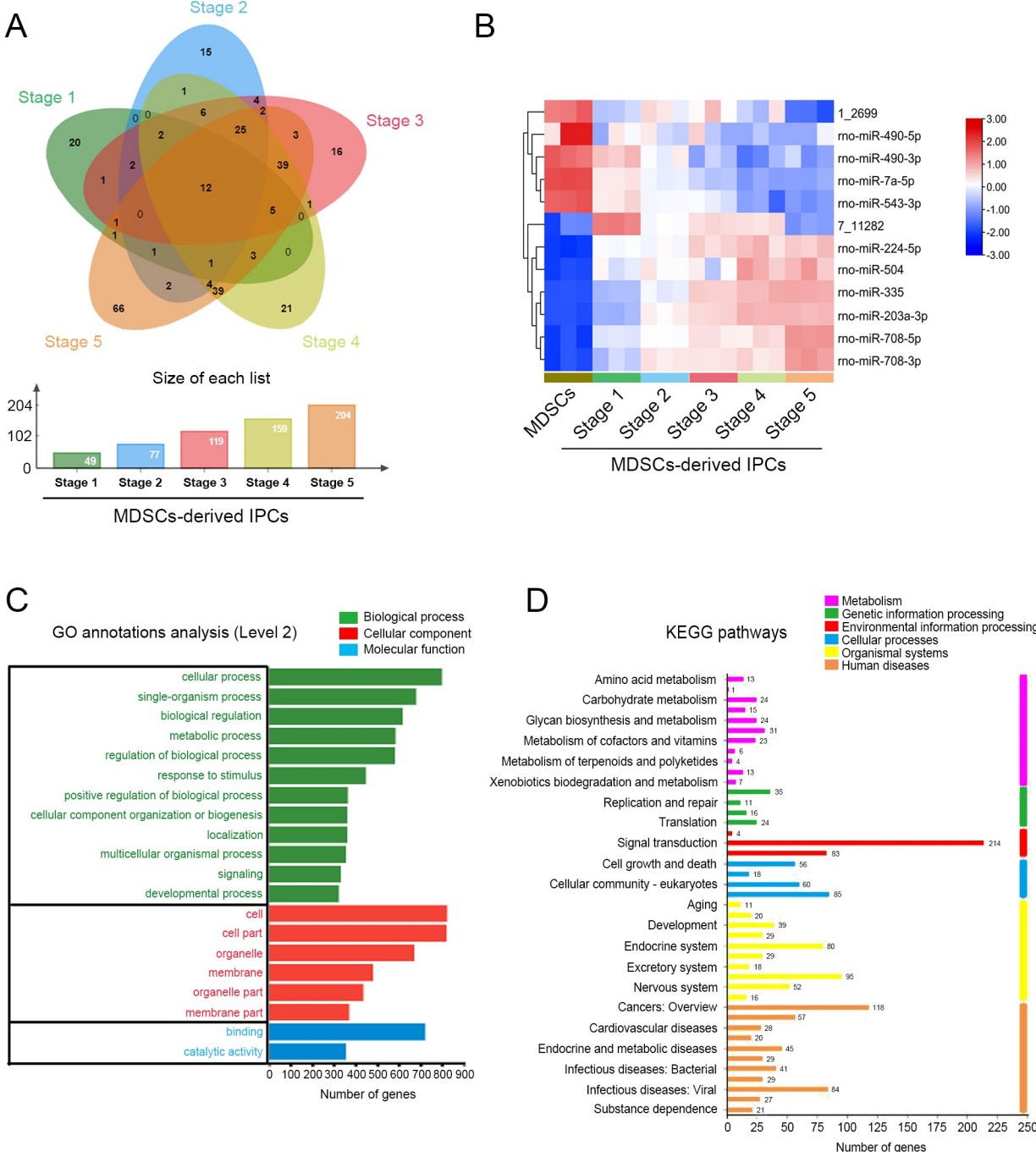

**Fig 3. Differentially expressed miRNAs (DEmiRNAs) profiling and bioinformatic analysis.** (A) The unique and common DEmiRNAs at stage 1–5 of MDSCs-derived IPCs differentiation were visualized by a Venn diagram. (B) The expression of 12 common DEmiRNAs from stage 1 to 5 was shown by a heatmap. (C) The functions of target genes of DEmiRNAs were enriched by gene ontology (GO) analysis. (D) The functions of target genes were enriched through KEGG pathway analysis.

shown in Fig 3B, the expression of rno-miR-490-5p, rno-miR-490-3p, rno-miR-7a-5p, and rno-miR-543-3p showed a decreasing trend from stage 1 to stage 5 of differentiation. In contrast, the expression of rno-miR-224-5p, rno-miR-504, rno-miR-335, rno-miR-203a-3p, rno-miR-708-5p, and rno-miR-708-3p presented an increasing trend. These DEmiRNAs were

considered as the potential miRNAs related to the regulation of MDSCs-derived IPCs differentiation.

### GO and KEGG enrichment analyses

To further investigate the roles of DEmiRNAs in the differentiation of MDSCs into IPCs, putative target genes were predicted using miRanda, TargetScan, and RNAhybrid databases. A total of 7145 target genes were predicted and then annotated using GO terms and KEGG pathways. GO function analysis showed that these target genes were mainly clustered to 12 terms for biological process, 6 for cellular component, and 2 for molecular function, respectively (Fig 3C). Most of these GO terms were closely related to cellular process and biological regulation that play important roles during IPCs differentiation. In addition, KEGG pathway analysis showed that there were 161, 86, 301, 219, 389, and 499 genes associated with metabolism, genetic information processing, environmental information processing, cellular processes, organismal systems, and human diseases, respectively (Fig 3D).

### Overexpression of miR-708-5p promoted the MDSCs-derived IPCs differentiation *in vitro*

According to the bioinformatics analysis, miR-708-5p draw our attention due to persistently increased expression trend from stage 1–5 of differentiation. To verify the expression of miR-708-5p during MDSCs-derived IPCs differentiation, qRT-PCR was performed and showed that the expression level of miR-708-5p was significantly increased from stage 1 to stage 5 in the progression of MDSCs-derived IPCs differentiation ($P < 0.01$) (Fig 4A). To further explore the specific function of miR-708-5p during differentiation, miR-708-5p was overexpressed through the transfection of miR-708-5p mimics into MDSCs. The overexpression efficiency of miR-708-5p mimics was confirmed by qRT-PCR, which showed that the expression of miR-708-5p in MDSCs transfected with miR-708-5p mimics was markedly higher than NC ($P < 0.05$) (Fig 4B). In addition, Pdx1, Ngn3, Nkx2.2, Nkx6.1, Gata4, Gata6, Pax4, and Pax6 are the pivotal transcription factors for early pancreatic development. The overexpression of miR-708-5p in MDSCs dramatically upregulated the expression of these transcription factors compared with NC cells ($P < 0.05$) (Fig 4C). Moreover, flow cytometry showed that the percentage of insulin significantly increased in MDSCs-derived IPCs of stage 5 in comparison to that in NC cells ($P < 0.001$). MDSCs overexpressed miR-708-5p presented more insulin positive ratio than MDSCs-derived IPCs ($P < 0.001$) (Fig 4D). Furthermore, immunofluorescence staining partially verified the above results of qRT-PCR and flow cytometry. The co-expression of insulin/C-peptide, insulin/Pdx1, and insulin/Nkx6.1 presented high expression in MDSCs overexpressed miR-708-5p (Fig 4E–4G).

### miR-708-5p inhibited STK4-mediated Hippo-YAP1 signaling pathway

To further elaborate on the molecular mechanism of miR-708-5p on MDSCs-derived IPCs differentiation, bioinformatics analysis was performed to predict the target genes of miR-708-5p. The results showed that miR-708-5p has 22 potential target genes. Among them, STK4 (a kind of conserved serine/threonine kinase) attracted our attention, due to its regulatory effect on cell proliferation and differentiation [20]. The binding site of miR-708-5p on STK4 was predicted by Starbase (Fig 5A). The interaction between miR-708-5p and STK4 was verified by a dual-luciferase reporter assay. The results showed that the co-transfection of miR-708-5p mimics and STK4-WT led to a significant decrease in luciferase activity compared to transfection with STK4-WT alone ($P < 0.01$). However, there were no obvious differences in luciferase activity when miR-708-5p mimics co-transfection with STK4-MT in comparison to

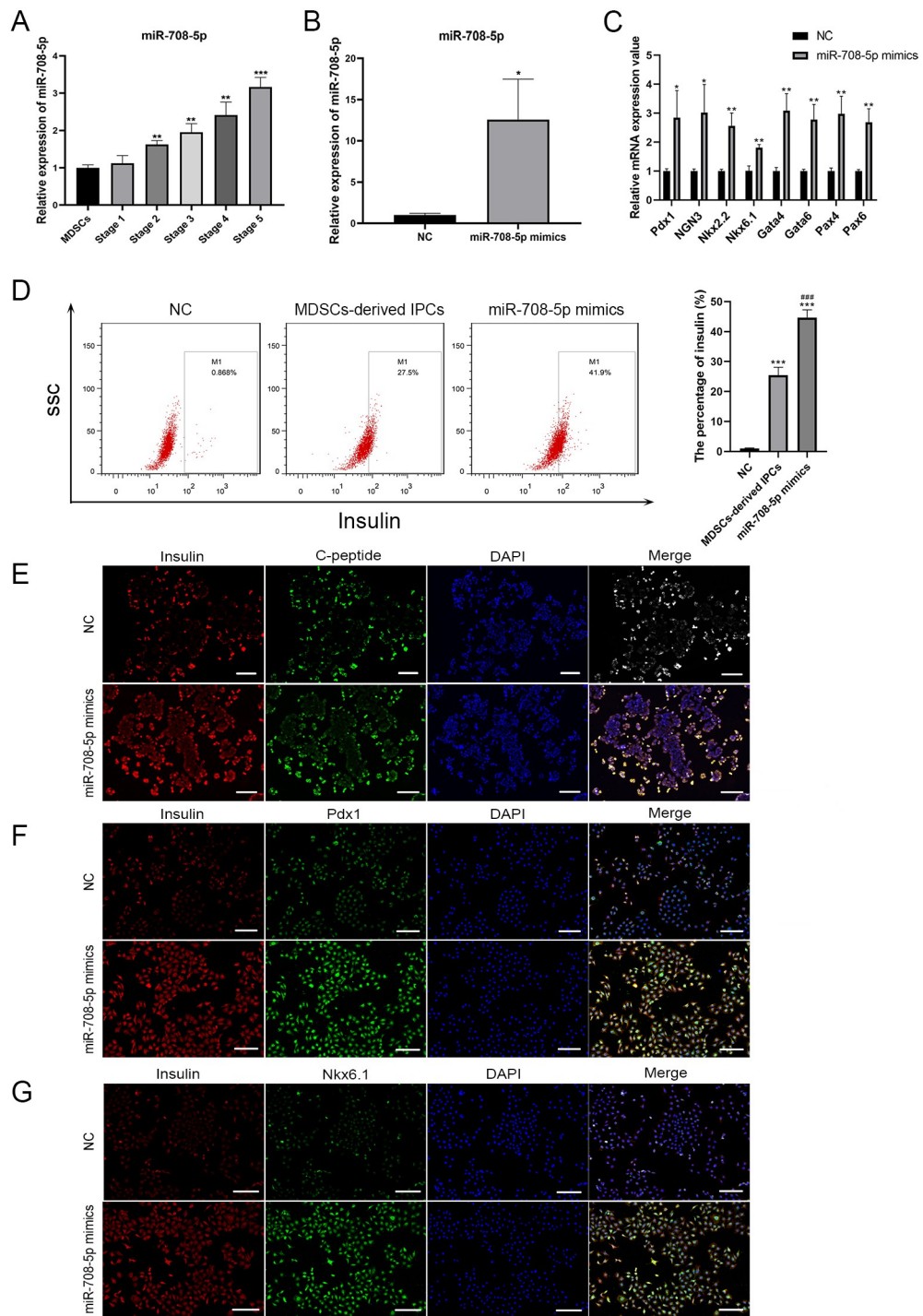

**Fig 4. Overexpression of miR-708-5p promoted the differentiation of MDSCs into IPCs.** (A) The expression of miR-708-5p in MDSCs and MDSCs-derived IPCs at stage 1–5 of differentiation was verified by qRT-PCR. $^{**}P < 0.01$, and $^{***}P < 0.001$ compared with MDSCs. (B) The expression of miR-708-5p in MDSCs overexpressed miR-708-5p (miR-708-5p mimics) was measured by qRT-PCR. $^{*}P < 0.05$ compared with the NC. (C) The expression levels of key transcription factors (Pdx1, NGN3, Nkx2.2, Nkx6.1, Gata4, Gata6, Pax4, and Pax6) in pancreatic β-cells were examined by qRT-PCR. $^{*}P < 0.05$ and $^{**}P < 0.01$ compared with the NC. (D) The percentage of insulin in MDSCs-derived IPCs with miR-708-5p overexpression was analyzed by flow cytometry. $^{***}P < 0.001$ compared with the NC and $^{\#\#\#}P < 0.001$ compared with the MDSCs-derived IPCs. (E-F) Co-immunostaining of insulin/C-peptide, insulin/Pdx1, and insulin/Nkx6.1 in MDSCs-derived IPCs with miR-708-5p overexpression. The nuclear was stained with DAPI in blue. Scale bar is 200 μm.

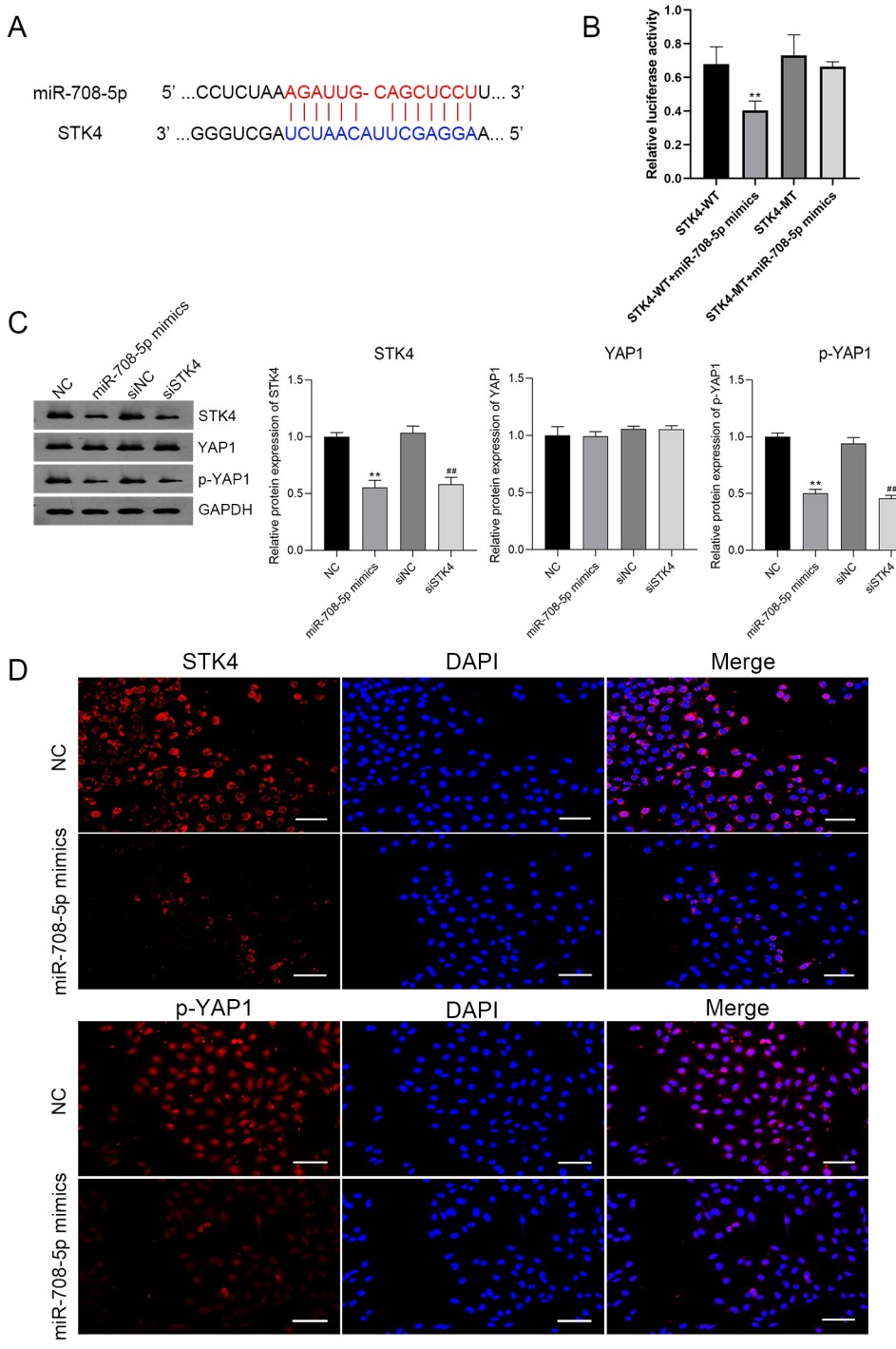

**Fig 5. miR-708-5p inhibited STK4-mediated Hippo-YAP1 signaling pathway.** (A) The targeting sequence of miR-708-5p in the 3'UTR of STK4 was predicated using Starbase (http://starbase.sysu.edu.cn/index.php). (B) The interaction between miR-708-5p and STK4 was identified using dual-luciferase reporter assay. $^{**}P < 0.01$ compared with the STK4-WT group. (C) The relative protein expression levels of STK4, YAP1, and p-YAP1 in MDSCs overexpressed miR-708-5p (miR-708-5p mimics) and silenced with STK4 (siSTK4) were examined using Western blot. $^{**}P < 0.01$ compared with the NC and $^{##}P < 0.01$ compared with the siRNA-negative control (siNC). (D) The expression of STK4 and p-YAP1 in MDSCs overexpressed miR-708-5p was verified by immunofluorescence staining. Nuclear DAPI staining was presented in blue. Scale bar is 200 μm.

transfection with STK4-MT alone (Fig 5B). In addition, STK4 is a pivotal kinase regulating the Hippo-YAP1 signaling pathway through the phosphorylation of YAP1. miR-708-5p overexpression dramatically reduced the protein expression of STK4 and phosphorylated YAP1 (p-YAP1) compared with the NC ($P < 0.01$). siSTK4 showed the same effect with miR-708-5p overexpression, evidenced by the significantly decreased expression of STK4 and p-YAP1 compared to siNC ($P < 0.01$). However, the protein expression of YAP1 did not show obvious differences among groups (Fig 5C). Furthermore, immunofluorescence staining confirmed the decreased expression of STK4 and p-YAP1 in MDSCs with miR-708-5p overexpression (Fig 5D).

## Discussion

Pancreatic β cell lose is a main characteristic of both type I and type II diabetes [21, 22]. Stem cells are a renewable source of pluripotent cells, which can be utilized to insulin-producing β cells repair and regeneration during diabetes treatment [23, 24]. In this study, we screened 12 DEmiRNAs involved in the differentiation of MDSCs into IPCs. Among them, miR-708-5p was able to promote IPCs differentiation probably by targeting STK4-mediated Hippo-YAP1 signaling pathway.

MDSCs are gradually becoming a favorable candidate for insulin-producing β cells regeneration, thereby treating diabetes. Wang et al. [25] revealed that bovine MDSCs had the potential to develop into insulin-secreting cells. Mitutsova et al. [6] proved that MDSCs had the ability of differentiating into mature insulin-expressing cells in pancreatic islets of diabetic rats. Consistent with previous studies, IPCs were successfully differentiated from MDSCs along with positive expression of insulin and C-peptide (an active form of insulin) in this study. These results suggested that MDSCs have the capacity of differentiating into IPCs, exhibiting a promising strategy for the treatment of diabetes.

Emerging data suggest that miRNAs play a pivotal role in IPCs differentiation from diverse stem cells [10]. Guo et al. indicated that the differentiation of induced pluripotent stem cells into IPCs is dependent on miRNA-related mechanisms [26]. MiRNA-101a and miRNA-107 showed prominent expression during the differentiation of adipose-derived mesenchymal stem cells into IPCs [27]. Williams et al. demonstrated that overexpressing specific miRNAs (such as miR-375 and miR-7) in islets contributes to achieve better differentiation of stem cells into IPCs [28]. These previous studies indicated that miRNAs act as a critical regulatory role in the differentiation of IPCs. In this study, miRNA microarray assay was performed to screen the DEmiRNAs involved in the differentiation of MDSCs into IPCs. There were 12 common DEmiRNAs obtained in the stage 1–5 of differentiation, suggesting that these DEmiRNAs may be involved in the regulation of MDSCs differentiation into IPCs. In order to further investigate the underlying mechanisms, the target genes of DEmiRNAs were predicted and their functions were enriched via GO and KEGG analyses. Enrichment analyses showed that these target genes were mainly related to cellular process and signal transduction, includes cell growth and death, apoptosis signaling pathway, insulin/IGF pathway-MAPK cascade, etc. These results indicate that DEmiRNAs may moderate MDSCs-derived IPCs differentiation through regulating target genes involving these processes.

miR-708-5p is a tumor suppressive miRNA involved in cell proliferation and differentiation [29, 30]. In this study, we found that miR-708-5p presented the increased expression during the differentiation of MDSCs into IPCs. In addition, miR-708-5p overexpression increased the expression of specific β-cell transcription factors and insulin positive ratio in MDSCs-derived IPCs. These results suggested that miR-708-5p has the ability of promoting the differentiation of MDSCs into IPCs. To further explore the downstream mechanism of miR-708-5p

regulating MDSCs-derived IPCs differentiation, STK4 was predicted and identified as a target gene of miR-708-5p. STK4 is a conserved serine/threonine kinase regulating cell proliferation and differentiation through the Hippo signaling pathway [31]. Previous studies have demonstrated that STK4 may exert a negative regulatory effect on IPCs differentiation via phosphorylating YAP1 in the Hippo-YAP1 signaling pathway [31, 32]. Our results showed that the expression of STK4 and p-YAP1 was downregulated in MDSCs overexpressed miR-708-5p. Therefore, we speculated that miR-708-5p may promote the differentiation of MDSCs into IPCs by targeting STK4-mediated Hippo-YAP1 signaling pathway.

In conclusion, 12 DEmiRNAs were closely related to the differentiation of MDSCs into IPCs. miR-708-5p played a key role in promoting the differentiation of MDSCs into IPCs probably via targeting STK4-mediated Hippo-YAP1 signaling pathway. This study not only throws light on the potential mechanism of miR-708-5p regulating the differentiation of MDSCs into IPCs, but also provides an important foundation for diabetes therapy. However, more data are needed to illuminate the underlying mechanisms of miR-708-5p/STK4/Hippo-YAP1 axis in MDSCs-derived IPCs differentiation. Besides, the functions of the remaining 11 DEmiRNAs are still need to be studied.

## Supporting information

**S1 Raw images.**
(PDF)

## Acknowledgments

MiRNA sequencing in this study was supported by Hangzhou Bio-science Co., Ltd, China.

## Author Contributions

**Conceptualization:** Yu Ren.

**Data curation:** Yu Ren, Xiao Wang, Hongyu Liang.

**Funding acquisition:** Yu Ren.

**Writing – original draft:** Yu Ren.

**Writing – review & editing:** Yuzhen Ma.

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
