## [Decision Letter · Decision Letter 0]

12 Jan 2022

PONE-D-21-35752Differentially expressed microRNAs during the differentiation of muscle-derived stem cells into insulin-producing cells, a promoting role of microRNA-708-5p/STK4 axisPLOS ONE

Dear Dr. Ma,

Thank you for submitting your manuscript to PLOS ONE. After careful consideration, we feel that it has merit but does not fully meet PLOS ONE’s publication criteria as it currently stands. Therefore, we invite you to submit a revised version of the manuscript that addresses the points raised during the review process.

As you will see, both referees highlight the potential interest of the findings. However, they have raised a number of concerns and suggestions to improve the manuscript, or to strengthen the data and the conclusions drawn. As the reports are below, I will not detail them here, as we think all points need to be addressed. Given the constructive referee comments, we would like to invite you to revise your manuscript with the understanding that both referee concerns must be addressed in the revised manuscript and in a point-by-point response. Acceptance of your manuscript will depend on a positive outcome of a second round of review.

We look forward to receiving your revised manuscript.

Kind regards,

Abdul Qadir Syed, PhD

Academic Editor

PLOS ONE

Journal Requirements:

Reviewers' comments:

Reviewer's Responses to Questions

**Comments to the Author**

1. Is the manuscript technically sound, and do the data support the conclusions?

Reviewer #1: Yes

Reviewer #2: No

2. Has the statistical analysis been performed appropriately and rigorously? 

Reviewer #1: Yes

Reviewer #2: Yes

3. Have the authors made all data underlying the findings in their manuscript fully available?

Reviewer #1: No

Reviewer #2: Yes

4. Is the manuscript presented in an intelligible fashion and written in standard English?

Reviewer #1: No

Reviewer #2: Yes

5. Review Comments to the Author

Reviewer #1: The objective of the study is well thought and executed.

But there are so many places in the manuscript which need to be improved.

There were no mention of which diabetes was studied here Type I or Type II?

Scientific English need to be used (centrifuging=centrifugation; 150 g=150xg) and reviewed carefully before submission. As line 224 mention: The separate and overlapping DEmiRNAs...... It may be refrain as "Unique and common" DEmiRNAs.... Last line mentioned "Left 11", it can be refrain as "remaining 11".

There are many places where sentences need to be reframed. Few sentences are repeated in Introduction and discussion. Every time Scale bar has been mentioned as separate sentence without making a sentence, which need to be corrected. p-values either be given in sentence or in parentheses.

What is RPMI? Is this a growth medium or Institute?

nictinamide=nicotinamide.

Version and references need to be provided for used Softwares and databases.

Methods: What were the analysis parameters used for processing of Raw sequence data and quality passed data?

Wild-type STK4 (STK4-WT) and the mutant 141 STK4 (STK4-MT) were cloned into pGL3 alkaline luciferase vector. Explain Mutation experiment.

Line 171: performed at least three independent repetitions. Is this assumption or confirmation regarding replicates?

Results:

Sequencing: No result has been mentioned about sequencing (Data obtained, processed in terms of reads and quality). pls provide this detail. What

Line 193: These two markers; Which are these two markers? Explain.

Line 208: Pls provide Figure number for heatmap.

There were 7145 target genes identified but only mention of 1655 for GO term analysis, what about other genes? Pls explain how these GO terms (12 biological process, 6 cellular components and 2 molecular functions) are related to IPCs differentiation.

Venn diagram and heatmap and bar charts are not visible, high quality figures may be submitted.

Line 311: Emerging data suggest that miRNAs act as a pivotal role in IPCs differentiation; "miRNAs play a pivotal role". Refrain the sentence.

Line 315: These researchers; Which researchers referred here?

Line 322: The results showed that these target genes were mainly related to cellular process and signal transduction. "No mention of Which cellular process and signal transduction." Explain and give details.

Line 334: STK4 exerts a regulatory effect on IPCs differentiation via phosphorylating YAP1....... Which kind of regulatory effect "Positive or negative"? Conclusion was based on this comparison and effect was not explained in the manuscript.

Reviewer #2: In the current study, the authors have tried to show the importance and involvement of miR-708-5p during differentiation of MDSCs into insulin producing cells. Although the study is important, experiments needs to be revised properly with proper controls.

1. Most of the experiments involve the expression of proteins by immunofluorescence studies. However, they lack proper control and comparisons. In Fig.1 the MDSCs marker expressions needs to be compared with the 0h cells. Likewise, in Fig.2, the insulin / c-peptide expression can be compared at each stage.

2. Similarly, the experiments involving addition of miR-708-5p mimic needs untreated cells as wells as NC-miRNA as controls.

3. The image quality needs to be improved throughout. The flow plots are not possible to read.

4. Authors need to explain about the STK4-Mut in detail. Is it having a mutated miR-708 binding site? If yes the details needs to be provided.

6. PLOS authors have the option to publish the peer review history of their article (what does this mean?). If published, this will include your full peer review and any attached files.

Reviewer #1: **Yes: **Ram Nageena Singh

Reviewer #2: No

---

## [Author Response · Author response to Decision Letter 0]

25 Feb 2022

Thanks for giving us the opportunity to submit a revised draft of the manuscript “Differentially expressed microRNAs during the differentiation of muscle-derived stem cells into insulin-producing cells, a promoting role of microRNA-708-5p/STK4 axis” for publication in the Journal of PLOS ONE. We appreciate the time and effort that editors and reviewers dedicated to providing feedback on our manuscript and are grateful for the insightful comments on and valuable improvements to our paper. We have incorporated most of the suggestions made by the reviewers. Those revisions are highlighted in the manuscript with tracked changes. Please see below, in blue, for a point-by-point response to the reviewers’ comments and concerns.

Reviewer #1: The objective of the study is well thought and executed.

But there are so many places in the manuscript which need to be improved.

There were no mention of which diabetes was studied here Type I or Type II?

R: This study is not specific to the type of diabetes. Stem cell therapy is a promising therapeutic approach for both Type I and Type II diabetes. Thus, the type of diabetes is not specified in this study.

Scientific English need to be used (centrifuging=centrifugation; 150 g=150xg) and reviewed carefully before submission. 

R: Thanks for pointing this out. We have reviewed this manuscript carefully and revised these non-standard writing.

As line 224 mention: The separate and overlapping DEmiRNAs...... It may be refrain as "Unique and common" DEmiRNAs.... Last line mentioned "Left 11", it can be refrain as "remaining 11".

R: Thanks for pointing this out. We have modified these words as you suggested.

There are many places where sentences need to be reframed. Few sentences are repeated in Introduction and discussion. 

R: Thanks for your suggestion. We have revised sentences that were repeated in the Discussion with the Introduction.

Every time Scale bar has been mentioned as separate sentence without making a sentence, which need to be corrected. p-values either be given in sentence or in parentheses.

R: Thanks for pointing this out. We have revised the “Scale bar = 100 or 200 μm” into a sentence “Scale bar is 100 or 200 μm” in the Figure legends. P-values have been given in sentence, e.g., “***P < 0.001 compared with the Control.”.

What is RPMI? Is this a growth medium or Institute?

R: RPMI is a growth medium and its full name has been modified to the Roswell Park Memorial Institute (RPMI) medium in the Methods section.

nictinamide=nicotinamide.

R: The word “nictinamide” has been revised to “nicotinamide”.

Version and references need to be provided for used Softwares and databases.

R: Thanks for pointing this out. The versions and references for used softwares and databases have been added to the Methods section.

Methods: What were the analysis parameters used for processing of Raw sequence data and quality passed data?

R: The high-quality sequencing data were screened according to the criteria as follows: 1) Remove the 3' linker sequence in the reads, and remove the reads without insert fragments due to the self-ligation of the linker; 2) Remove the reads with low sequencing quality in 3'-base (the quality value is less than 20); 3) Remove reads containing unknown base N; 4) Choose reads with length between 18nt and 32nt. These analysis parameters have been added to the Methods section.

Wild-type STK4 (STK4-WT) and the mutant 141 STK4 (STK4-MT) were cloned into pGL3 alkaline luciferase vector. Explain Mutation experiment.

R: STK4-MT was established via mutating the putative binding site of miR-708-5p in STK4 3’-UTR, which has been described in the Methods section.

Line 171: performed at least three independent repetitions. Is this assumption or confirmation regarding replicates?

R: The replicate determination is to confirm the results. 

Results:

Sequencing: No result has been mentioned about sequencing (Data obtained, processed in terms of reads and quality). pls provide this detail. What

R: Thanks for pointing this out. The data acquisition and processing about sequencing have been provided in the Methods section. In the Results section, we mainly described the obtained differentially expressed miRNAs from sequencing.

Line 193: These two markers; Which are these two markers? Explain.

R: These two markers are insulin and C-peptide that have been explained in the Results section.

Line 208: Pls provide Figure number for heatmap.

R: The heatmap is in Figure 3B, which has been provided in the Results section.

There were 7145 target genes identified but only mention of 1655 for GO term analysis, what about other genes? Pls explain how these GO terms (12 biological process, 6 cellular components and 2 molecular functions) are related to IPCs differentiation.

R: In this study, we mainly explored the KEGG pathways involved in metabolism, genetic information processing, environmental information processing, cellular processes, organismal systems, and human diseases. A total of 1655 genes were enriched in these pathways, therefore, we mentioned 1655 target genes for KEGG pathway analysis in this manuscript. For GO terms analysis, most of GO terms were closely related to cellular process and biological regulation that play important roles during IPCs differentiation. The association between GO terms and IPCs differentiation has been explained in the Results section.

Venn diagram and heatmap and bar charts are not visible, high quality figures may be submitted.

R: Thanks for pointing this out. We have re-provided these images with high resolution (300 dpi).

Line 311: Emerging data suggest that miRNAs act as a pivotal role in IPCs differentiation; "miRNAs play a pivotal role". Refrain the sentence.

R: We have revised the “miRNAs act as a pivotal role” to "miRNAs play a pivotal role" in the Discussion section as you suggest.

Line 315: These researchers; Which researchers referred here?

R: “These researchers” means the previous studies mentioned in the manuscript. We have modified “These researchers” to “These previous studies” in the Discussion section.

Line 322: The results showed that these target genes were mainly related to cellular process and signal transduction. "No mention of Which cellular process and signal transduction." Explain and give details.

R: Cellular process and signal transduction includes cell growth and death, apoptosis signaling pathway, insulin/IGF pathway-MAPK cascade, etc. That has been added in the Discussion section.

Line 334: STK4 exerts a regulatory effect on IPCs differentiation via phosphorylating YAP1....... Which kind of regulatory effect "Positive or negative"? Conclusion was based on this comparison and effect was not explained in the manuscript.

R: STK4 may exert a negative regulatory effect on IPCs differentiation, which has been revised in the Discussion section. There is direct evidence for the effect of STK4 on IPCs differentiation in this study, which will be further confirmed in subsequent investigation. In this study, we mainly focus on the role of miR-708-5p in IPCs differentiation.

Reviewer #2: In the current study, the authors have tried to show the importance and involvement of miR-708-5p during differentiation of MDSCs into insulin producing cells. Although the study is important, experiments needs to be revised properly with proper controls.

1. Most of the experiments involve the expression of proteins by immunofluorescence studies. However, they lack proper control and comparisons. In Fig.1 the MDSCs marker expressions needs to be compared with the 0h cells. Likewise, in Fig.2, the insulin / c-peptide expression can be compared at each stage.

R: Thanks for your professional suggestion. The experimental design of this study referred to the previous research by Xu et al. (2019) (PMID: 30767782). The comparisons of the results in Fig. 1 have been revised as you suggest. In Fig. 2, the insulin/C-peptide expression of MDSCs-derived IPCs was compared to MDSCs without differentiation. 

2. Similarly, the experiments involving addition of miR-708-5p mimic needs untreated cells as wells as NC-miRNA as controls.

R: Thanks for your professional suggestion. The experimental design of this study referred to the previous research by Xu et al. (2019) (PMID: 30767782), which can provide enough evidence to confirm the effect of miR-708-5p.

3. The image quality needs to be improved throughout. The flow plots are not possible to read.

R: Thanks for pointing this out. We have improved images with high resolution.

4. Authors need to explain about the STK4-Mut in detail. Is it having a mutated miR-708 binding site? If yes the details needs to be provided.

R: STK4-MT was established via mutating the putative binding site of miR-708-5p in STK4 3’-UTR, which has been described in the Methods section.

---

## [Decision Letter · Decision Letter 1]

21 Mar 2022

PONE-D-21-35752R1Differentially expressed microRNAs during the differentiation of muscle-derived stem cells into insulin-producing cells, a promoting role of microRNA-708-5p/STK4 axisPLOS ONE

Dear Dr. Ma,

Thank you for submitting your manuscript to PLOS ONE. After careful consideration, we feel that it has merit but does not fully meet PLOS ONE’s publication criteria as it currently stands. Therefore, we invite you to submit a revised version of the manuscript that addresses the points raised during the review process.

Specifically, there is still one comment which was not addressed in the revised manuscript. I highly suggest that author should address this comment in the final version of manuscript. 

We look forward to receiving your revised manuscript.

Kind regards,

Abdul Qadir Syed, PhD

Academic Editor

PLOS ONE

Journal Requirements:

Additional Staff Editor Comments (if provided): PLOS ONE does not provide copyediting or proofs of accepted manuscripts. We therefore recommend that you carefully review your manuscript and correct any language errors at this time.

Reviewers' comments:

Reviewer's Responses to Questions

**Comments to the Author**

1. If the authors have adequately addressed your comments raised in a previous round of review and you feel that this manuscript is now acceptable for publication, you may indicate that here to bypass the “Comments to the Author” section, enter your conflict of interest statement in the “Confidential to Editor” section, and submit your "Accept" recommendation.

Reviewer #1: (No Response)

Reviewer #2: All comments have been addressed

2. Is the manuscript technically sound, and do the data support the conclusions?

Reviewer #1: Yes

Reviewer #2: Yes

3. Has the statistical analysis been performed appropriately and rigorously? 

Reviewer #1: Yes

Reviewer #2: Yes

4. Have the authors made all data underlying the findings in their manuscript fully available?

Reviewer #1: Yes

Reviewer #2: Yes

5. Is the manuscript presented in an intelligible fashion and written in standard English?

Reviewer #1: Yes

Reviewer #2: Yes

6. Review Comments to the Author

Reviewer #1: There is still one comment which need to be addressed.

Statistical Analysis:

Line 182: Each experiment was performed at least three independent repetitions.

This sentence reflects that Authors were confirmed about number of replicates. Why "at least"? Why not that "All the experiments were performed with 3 replicates"?

Pls revise the sentence.

Reviewer #2: The authors have answered all the comments satisfactorily. Now, it can be considered for publication.

7. PLOS authors have the option to publish the peer review history of their article (what does this mean?). If published, this will include your full peer review and any attached files.

Reviewer #1: **Yes: **Ram Nageena Singh

Reviewer #2: No

---

## [Author Response · Author response to Decision Letter 1]

22 Mar 2022

Thanks for giving us the opportunity to submit a revised draft of the manuscript “Differentially expressed microRNAs during the differentiation of muscle-derived stem cells into insulin-producing cells, a promoting role of microRNA-708-5p/STK4 axis” for publication in the Journal of PLOS ONE. We appreciate the time and effort that editors and reviewers dedicated to providing feedback on our manuscript and are grateful for the insightful comments on and valuable improvements to our paper. We have incorporated most of the suggestions made by the reviewers. Those revisions are highlighted in the manuscript with tracked changes. Please see below, in blue, for a point-by-point response to the reviewers’ comments and concerns.

Journal Requirements:

R: We have carefully reviewed the reference cited in this manuscript, and ensure that it is complete and correct.

Additional Staff Editor Comments (if provided): PLOS ONE does not provide copyediting or proofs of accepted manuscripts. We therefore recommend that you carefully review your manuscript and correct any language errors at this time.

R: We have carefully reviewed this manuscript, and corrected language errors.

Reviewer #1: There is still one comment which need to be addressed.

Statistical Analysis:

Line 182: Each experiment was performed at least three independent repetitions.

This sentence reflects that Authors were confirmed about number of replicates. Why "at least"? Why not that "All the experiments were performed with 3 replicates"?

Pls revise the sentence.

R: Thanks for pointing this out. We have revised the sentence “Each experiment was performed at least three independent repetitions.” to “All the experiments were performed with three independent repetitions.” in line 181.

Reviewer #2: The authors have answered all the comments satisfactorily. Now, it can be considered for publication.

R: Authors appreciate reviewer’s valuable comments and approval for publication.

---

## [Editor Report · Decision Letter 2]

24 Mar 2022

Differentially expressed microRNAs during the differentiation of muscle-derived stem cells into insulin-producing cells, a promoting role of microRNA-708-5p/STK4 axis

PONE-D-21-35752R2

Dear Dr. Ma, 

We’re pleased to inform you that your manuscript has been judged scientifically suitable for publication and will be formally accepted for publication once it meets all outstanding technical requirements.

Kind regards,

Abdul Qadir Syed, PhD

Academic Editor

PLOS ONE

---

## [Editor Report · Acceptance letter]

30 Mar 2022

PONE-D-21-35752R2 

Differentially expressed microRNAs during the differentiation of muscle-derived stem cells into insulin-producing cells, a promoting role of microRNA-708-5p/STK4 axis 

Dear Dr. Ma:

I'm pleased to inform you that your manuscript has been deemed suitable for publication in PLOS ONE. Congratulations! Your manuscript is now with our production department. 

Kind regards, 

on behalf of

Dr. Abdul Qadir Syed 

Academic Editor

PLOS ONE